TECHNICAL RELEASE

# Molecular Property Diagnostic Suite for COVID-19 (MPDS^COVID-19): an open-source disease-specific drug discovery portal

Lipsa Priyadarsinee[1,2], Esther Jamir[1], Selvaraman Nagamani[1,2], Hridoy Jyoti Mahanta[1,2], Nandan Kumar[1], Lijo John[1], Himakshi Sarma[1], Asheesh Kumar[1], Anamika Singh Gaur[3], Rosaleen Sahoo[1,2], S. Vaikundamani[1], N. Arul Murugan[4], U. Deva Priyakumar[5], G. P. S. Raghava[4], Prasad V. Bharatam[6], Ramakrishnan Parthasarathi[2,3], V. Subramanian[7], G. Madhavi Sastry[8] and G. Narahari Sastry[1,2,9,*,†]

1 CSIR–North East Institute of Science and Technology, Jorhat, 785006, India
2 Academy of Scientific and Innovative Research (AcSIR), Ghaziabad, 201002, India
3 CSIR-Indian Institute of Toxicology Research, Lucknow, 226001, Uttar Pradesh, India
4 Indraprastha Institute of Information Technology, Delhi, 110020, India
5 International Institute of Information Technology, Gachibowli, Hyderabad, 500032, India
6 National Institute of Pharmaceutical Education and Research, S.A.S. Nagar (Mohali), 160062, India
7 Department of Chemistry, Indian Institute of Technology, Chennai, 600036, India
8 Schrödinger Inc., Octave, Salarpuria Sattva Knowledge City, 1st Floor, Unit 3A, Hyderabad, 500081, India
9 Indian Institute of Technology (IIT) Hyderabad, Kandi, Sangareddy, Telangana, 502284, India

**Submitted:** 18 October 2023

* Corresponding author. E-mail: gnsastry@gmail.com; gnsastry@bt.iith.ac.in

† Current address: G. Narahari Sastry, Indian Institute of Technology (IIT), Hyderabad, Kandi, Sangareddy, Telangana, 502284, India.

Preprint submitted at https://doi.org/10.1101/2023.08.29.555437

## ABSTRACT

Molecular Property Diagnostic Suite (MPDS) was conceived and developed as an open-source disease-specific web portal based on Galaxy. MPDS^COVID-19 was developed for COVID-19 as a one-stop solution for drug discovery research. Galaxy platforms enable the creation of customized workflows connecting various modules in the web server. The architecture of MPDS^COVID-19 effectively employs Galaxy v22.04 features, which are ported on CentOS 7.8 and Python 3.7. MPDS^COVID-19 provides significant updates and the addition of several new tools updated after six years. Tools developed by our group in Perl/Python and open-source tools are collated and integrated into MPDS^COVID-19 using XML scripts. Our MPDS suite aims to facilitate transparent and open innovation. This approach significantly helps bring inclusiveness in the community while promoting free access and participation in software development.

**Availability & Implementation:** The MPDS^COVID-19 portal can be accessed at https://mpds.neist.res.in:8085/.

**Subjects** Software and Workflows, Bioinformatics, Cheminformatics

## STATEMENT OF NEED

### Background

In recent years, the world has witnessed the coronavirus disease-19 (COVID-19) caused by Severe Acute Respiratory Syndrome-CoronaVirus-2 (SARS-CoV-2). The SARS-CoV-2 infection resulted in a pandemic and caused severe damage to global healthcare and the economy [1, 2]. In developing new therapeutics and drugs for SARS-CoV-2, scientists have made

substantial progress in deciphering the pathophysiology of the disease and its prognosis. Important clues at the molecular level on viral entry, transcription, translation, translocation, and interaction with host cell receptors were obtained, providing definitive knowledge of the druggable target protein of the viral genome.

Molecular Property Diagnostic Suite (MPDS) is a Galaxy-based (RRID:SCR_006281) [3–5] open-source computational platform for drug discovery. Galaxy is ideal for developing web portals in diverse areas, such as bioinformatics, cheminformatics, genomics, proteomics, and Computer Aided Drug Design (CADD) analysis. MPDS results from our interest in developing open-source disease-specific web portals [6–8]. Galaxy offers the flexibility for users to deploy tools using multiple programming languages. Computational methods have played a significant role in drug discovery, involving finding hits, converting them into leads, and optimising them. Various computational methods have been used for target identification and validation, virtual screening, drug design and optimization, drug–drug interactions, personalized medicines, as well as Absorption, Distribution, Metabolism, Excretion, and Toxicity (or ADME-Tox) prediction. Computational approaches were crucial during pandemics or large-scale infections, as evidenced by the HIV drug discovery in the 90s and the COVID-19 drug and vaccine discovery during the recent pandemic. In this manuscript, we present a disease-specific web portal on COVID-19, which aligns with our earlier web portals, namely MPDS$^{TB}$ for tuberculosis [6] and MPDS$^{DM}$ for diabetes mellitus [7].

Developing customized computational tools and software is essential to comply with recent advances in computer science and data analytics. Considering the enormous differences among diseases, each disease may have specific requirements for (computational) drug design and discovery [3, 4]. Most existing tools focus on selected generic methods that can be applied to any disease. However, the MPDS disease-specific web portal aims to customize and integrate tools and methods for a specific disease, such as COVID-19. Drug discovery tools, such as Open Source Drug Discovery [9], Open Drug Discovery Toolkit [10], Supercomputing Facility for Bioinformatics and Computational Biology [11], CYCLICA [12], Orion-cloud-based platform [13], and Groningen Machine for Chemical Simulations [14], are widely used for drug discovery. Several databases/web servers have also been developed focusing on COVID-19 (Table S1 in GitHub [15]).

MPDS$^{COVID-19}$ is a customized web portal (Figure 1) designed for COVID-19 computational drug discovery. Thus, MPDS integrates many modules, including some specific to COVID-19. Galaxy provides access to various tools and vast amounts of data that can be integrated from different fields, such as bioinformatics, cheminformatics, genomics, and transcriptomics, and used to develop models based on advanced Artificial Intelligence (AI) and Machine Learning (ML) approaches [3–5, 16–18]. The advantage of this platform is that the source code is publicly available and users can customize and scale up the platform by integrating multiple tools and designing their workflows. The transparency, reproducibility, and accessibility of this platform made the developers integrate their scripts, tools, and codes efficiently. A list of critical features of the MPDS$^{COVID-19}$ portal, along with a brief justification, is provided below.

### *Integration*

This portal effectively integrates molecular modelling, informatics, simulations, literature surveys, and prior art on the disease's pathophysiology and drug discovery.



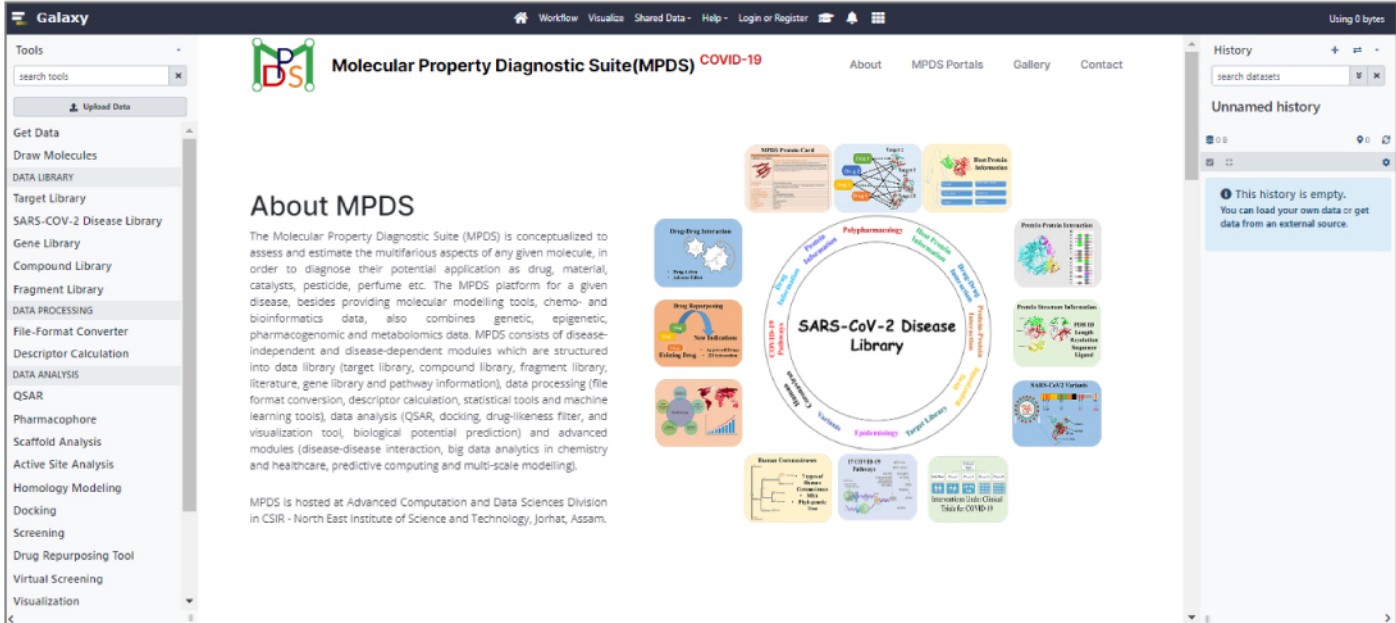

**Figure 1.** The home page of MPDS[COVID-19]. The portal can be accessed at https://mpds.neist.res.in:8085. The MPDS[COVID-19] has been structured in *Data library, Data processing, Data analysis, and Advanced modules.*

### Latest literature and updates

An important element of disease-specific web portals is access to the most recent information on drug development, related literature, and information on the disease. Also, any *in silico* script, algorithm, database, web portal, or useful software can easily be integrated with MPDS[COVID-19].

### Easy access to data

The web portal contains both disease-dependent and disease-independent modules, making it easy to access the data of a particular disease and perform analyses using the available tools.

### Data collection

The information is collected from known databases/servers such as Protein Data Bank (PDB) [19], PubMed [20], UniProt [21], DrugBank [22], DrugCentral [23], World Health Organization (WHO) [24], and other open access databases (Supplementary Table S2 in GitHub repository [15]), which make it easy for users to follow-up. These data have been segregated into three libraries – gene library, target library, and SARS-CoV-2 disease library in MPDS[COVID-19].

### Data modality

The MPDS[COVID-19] can handle different types of data to perform drug design calculations. These include structural data (2 Dimensional (2D) and 3 Dimensional (3D) structures of small and macro molecules in different file formats), chemical data (structures, chemical properties of compounds, molecular descriptors, and properties), and biological data,

including genetic and biological information such as DNA and protein sequences. The data library module of MPDS$^{COVID-19}$ uses and handles different types of data modalities, including structured text data (i.e., Comma Separated Value, Extensible Markup Language (XML), and JavaScript Object Notation files), structured data stored in databases, spreadsheets or tabular formats, categorical data with categories and labels (often used in ML calculations).

### Interface

This suite has the advantage of a user-friendly interface. Galaxy allows users to modify and add any desired tools or data with the developer's permission.

### Uniqueness

This portal allows to do computational drug discovery. It provides state-of-the-art computational tools as open access and the opportunity to engage in community-driven software development. MPDS$^{COVID-19}$ is an efficient platform for conducting virtual screening for COVID-19 associated targets, pharmacophore modelling, and developing ML models and 2D Quantitative Structure Activity Relationship (QSAR).

Approaches employed in drug discovery, especially in *in silico* methods, are witnessing tremendous changes owing to the unprecedented progress in hardware, software, AI, and Internet of Things (IoT)-based methods (Table S3 in GitHub [15]) [25]. Developing a disease-specific holistic portal and integrating existing knowledge with computational modelling holds promise in challenging the existing paradigms of computational drug discovery. To this end, our attempts to develop a series of disease-specific MPDS web portals may allow the research community to share their program and software freely.

## IMPLEMENTATION

### Our approach

Galaxy [4, 26] is a web-based application that can be used to create user-defined workflows and is readily intended to interact with any other software or script. The platform has been effectively used by scientists worldwide to analyse the biomedical data available in various fields of genomics, proteomics, transcriptomics, and bioinformatics (Figure S2 in GitHub [15]). We chose Galaxy because our work has been focused on various aspects, such as data availability, tools integration, reproducibility, and transparency for reuse. The Galaxy platform consists of four main components: (a) the main public Galaxy server [27]; (b) the Galaxy framework and software ecosystem [28]; (c) the Galaxy toolshed; and (d) the Galaxy community. The available servers (4), web portals (23), and tools (8) deployed using the Galaxy platform are listed in Table S4 in GitHub [15], along with their Uniform Resource Locator and use. The detailed information and architecture of Galaxy core components are stated in Figure 2.

### Architecture of the MPDS$^{COVID-19}$ software suite

Galaxy application can be installed on UNIX and Linux operating systems, and it has both frontend and backend with stability and flexibility in tools integration and data analysis. The backend of Galaxy is operated by a Python-based server with flexibility and is also driven by plugins. The frontend architecture makes the users' tasks easy and consistent,

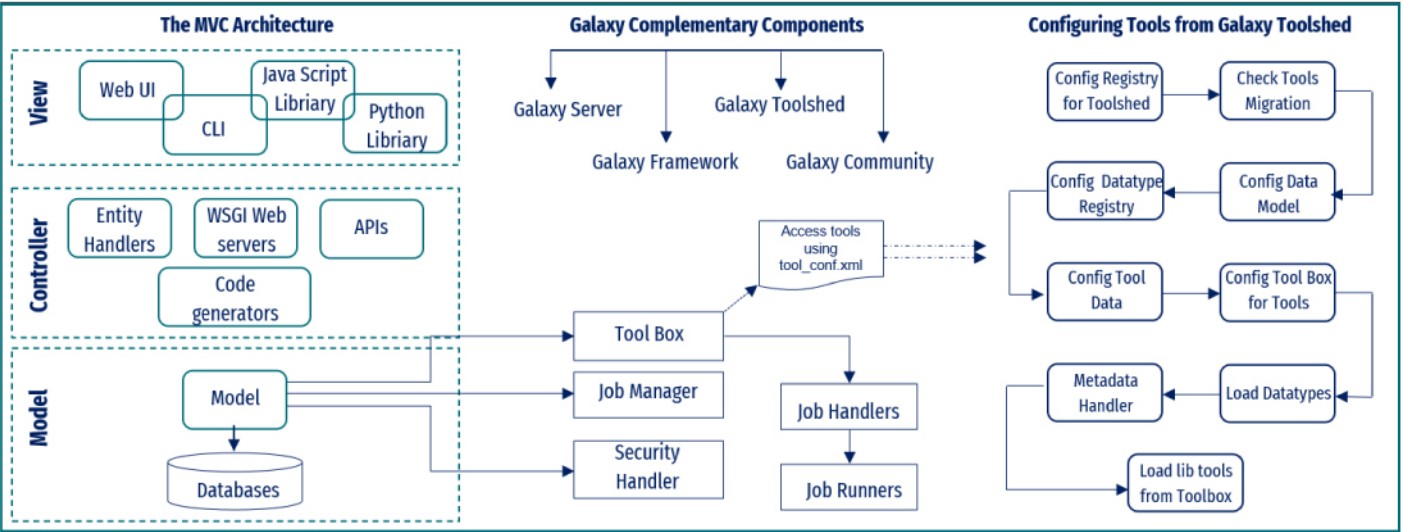

**Figure 2.** The architecture of Galaxy core components and tool configuration.

while the backend works more towards integrating different new technologies [5]. MPDS has been physically installed on one local server with four Quad Core processors and 160 cores. All the individual portals of MPDS are ported on virtual machines using 20 cores from the physical server. The administrator can effectively manage the user-submitted jobs through the admin panel in the portal. In the MPDS[COVID-19] web portal, the database on COVID-19 has been developed using an in-house Hypertext Preprocessor and MySQL scripts in the backend, while new tools and software were integrated using in-house Perl and Python scripts. The detailed architecture of the MPDS[COVID-19] is explained in Figure 3.

The design of the MPDS web portal involves three sections: (a) Data library, (b) Data processing, and (c) Data analysis. The data library section consists of disease-specific modules and provides information on SARS-CoV-2 (Figure 4). Thus, MPDS[COVID-19] has many modules related to SARS-CoV-2, and the web portal also provides links to tools specifically designed for antivirals in general and SARS-CoV-2 in particular. The major updates in the data library include the fragment library [29] as a separate module and a significantly revised form of the compound library [30]. The data processing module underwent little change except for the newer versions of the file format converter and descriptor calculation tool. However, substantial additions were made to the data analysis module. An attractive feature of the MPDS[COVID-19] is the addition of advanced modules employing ML approaches.

## Collection of data

The extensive literature survey from PubMed and Google Scholar and the use of freely accessible databases and tools collates detailed information on the literature, SARS-CoV-2 targets, genes and their pathway information. Additionally, it also contains information on repurposed drugs against various targets of SARS-CoV-2, mutational variants, polypharmacology for COVID-19 (Figure S3 in GitHub [15]) [31, 32], drug–drug interaction

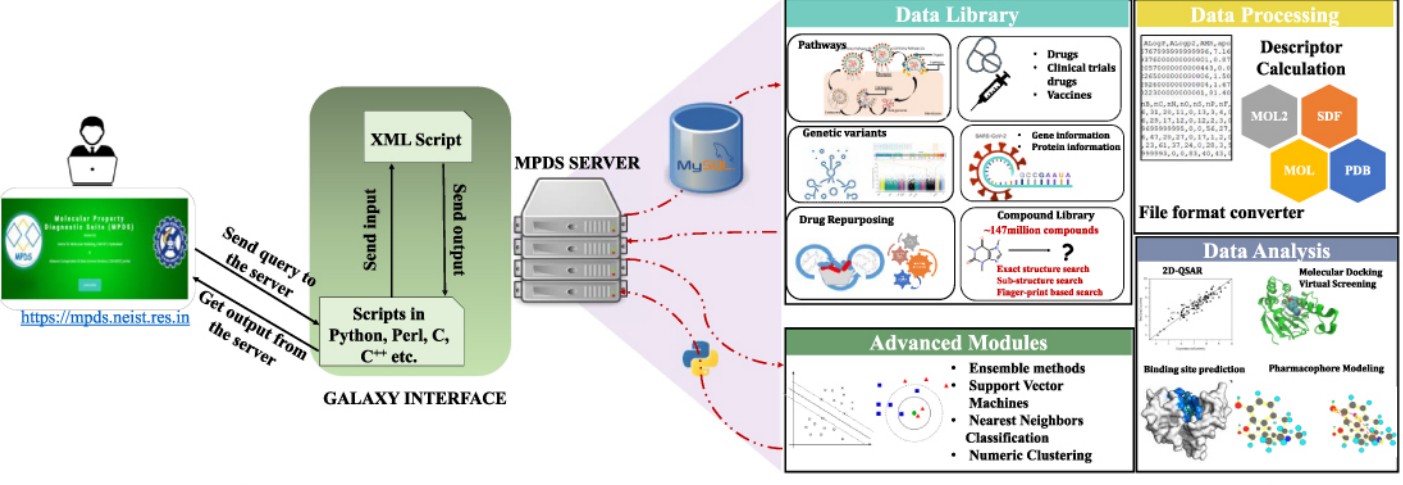

**Figure 3.** The Galaxy architecture of MPDS<sup>COVID-19</sup>. The users can upload their data (i.e., macro and small molecules) or select from the MPDS data library. The uploaded or selected information will be available in the Galaxy history panel. The users can submit the calculation in the main window, and the results will appear in the history panel.

information, Protein–Protein Interaction (PPI) [33, 34], host protein information, epidemiology, and inhibitors databases.

The extensive information on SARS-CoV-2 proteins was collected from the National Center for Biotechnology Information (NCBI) [35], PDB [19], PubMed [20], and Kyoto Encyclopaedia of Genes and Genomes (KEGG) [36]. The structure information was collected from PDB using the advanced search query option, then creating a custom report to get the desired information on each protein. PPI information was collected from various literature downloaded from PubMed and online searches using different keywords, like "PPI SARS-CoV-2 during viral entry", "PPI During Viral Propagation/survival", "PPI of Immune evasion", and other detailed analyses on PPI carried out in our group. Approved drugs and drugs in different phases of clinical trials were collected from databases such as ChEMBL [37], DrugBank [22], and PubChem [38]. Variants' epidemiology information was collected from the WHO [39] and the European Centre for Disease Prevention and Control [40]. The gene information contains the sequence, PDB identifier (ID), structure, domain information, active site residues and their function, along with various database IDs for structural, non-structural, and accessory proteins involved in SARS-CoV-2 [41]. The library also provides a detailed description of the pathways involved in COVID-19 infection. It contains information on nine PPIs that participate in various pathways, from viral entry, replication, and transcription to immune invasion [31, 32]. In the drug information, a list of Food and Drug Administration approved drugs and other emergency and promising drugs for COVID-19 are provided. This information gives an overview of all the drugs available for studying the development process of drugs against SARS-CoV-2.

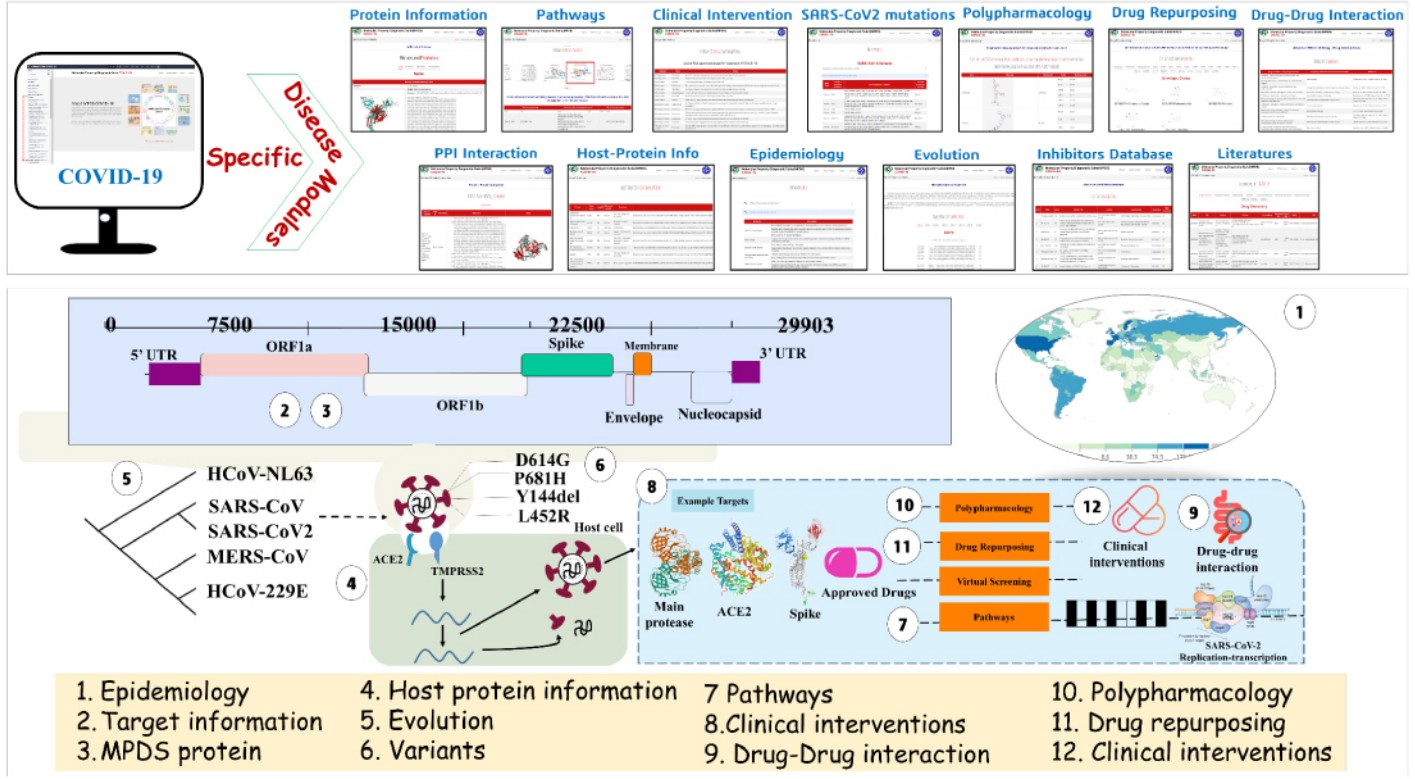

**Figure 4.** The diagrammatic representation depicting various disease-specific information incorporated in the MPDS^COVID-19 disease library.

## RESULTS

### Development of a disease-specific web portal

The MPDS disease-specific portal was developed on the Galaxy platform and provides easy access to a single platform of essential information to users worldwide. The web portal was designed into three modules: (1) Data Library, (2) Data Processing, and (3) Data Analysis.

### Disease-dependent modules

The Data library comprises different categories (target library, SARS-CoV-2 disease library, and gene library). The target library contains around 3,161 PDBs of proteins associated with SARS-CoV-2, with detailed structural information on their molecular weight, structure title, PDB IDs, experimental models, date, resolution, structure title, structure keywords, ligand name, and molecular weight.

### Preparation of the SARS-CoV-2 disease library

The literature module provides vast information on the COVID-19 pandemic outbreak, druggable targets, epidemiology studies in various regions, and geographical mapping of coronavirus disease worldwide. In these modules, the portal contains the libraries described in previous sections. The protein information contains sequence and structural data related to four structural, 16 non-structural, and nine accessory proteins of SARS-CoV-2. Importantly, users can access protein information, such as the sequence, 3D structure, functions, amino acid length, molecular weight, protein domain information, active site



residues, selected PDB ID, NCBI ID, UniProt ID, PUBMED ID for literature, and KEGG pathway ID. Information on different mutational variants of SARS-CoV-2, such as Variants Of Interest (VOIs), Variants Of Concern (VOCs), previously circulating VOIs, previously circulating VOCs, and Formerly Mentioned Variants for spike protein, and the epidemiology information are incorporated in the disease library.

The MPDS$^{COVID-19}$ also has 3D structures of the SARS-CoV-2 targets and inhibitors database for the respective targets. Other than this, the clinical trials dataset, cell experiment dataset, and virtual screening dataset are also available in this section. These datasets were collected and curated from the published literature. The database contains inhibitors along with the binding affinity from ChEMBL [37] and the Binding database [42]. Users can download the 3D structures of the 7,721 molecules for different computational drug discovery calculations. The visual representation of the SARS-CoV-2 disease library is illustrated in Figure 4.

### Gene library

The gene library in MPDS$^{COVID-19}$ contains a unique identifier called gene ID. Upon searching for a gene ID, the corresponding gene card is provided, containing the gene name, description, and characterization.

### Disease-independent modules

In addition to the data library module in the disease-dependent category, there are several disease-independent modules, as described in the following sections.

### Compound and fragment library

The compound library module in the initial MPDS suite had about 110 million compounds, and the chemical space was divided into classes. Each class represents a unique structural feature and, to limit the number of compounds in a group, the number of molecules in a class was divided into clusters. The recent version of the MPDS compound library [30] has over 149 million unique molecules and is distributed into 56 classes. Every molecule has an MPDS-AadharID, which unambiguously characterizes the molecule by assigning a unique class and cluster number [30]. Users can submit the data in a variety of forms, including Simplified Molecular-Input Line-Entry System (SMILES), International Union of Pure and Applied Chemistry (IUPAC) International Chemical Identifier (InChI), .mol, .mol2, and .sdf formats. If the query molecule exists in the database, an MPDS-AadharID is generated. If the query molecule is unavailable in the database, a message is displayed stating that the molecule does not exist in the database. In such a case, the user may contact the web server manager and a new MPDS-AadharID number will be generated. Recently, the fragment library module of the MPDS has been added. The fragment library contains 107,614 unique fragments, which may be further classified as rings, linkers, and substituents [29].

### Data processing

Different tools have unique requirements for the input files and produce the output files in another format. In order to fulfil the requirements of different tools, various file format converter tools have been used from the Galaxy toolshed for inter-conversion among various file formats. Input/output formats include viz., .sdf,. mol,. mol2,. cml, InChI, SMILES, and .pdb., and correspondingly, the output formats .mol2, .sdf, .mol, and .pdb are

incorporated into the MPDS. The module also includes a tool for converting the 2D structure of a compound into its 3D structure. MPDS provides two descriptor calculator tools, namely PaDEL (RRID:SCR_014272) and Chemistry Development Kit (RRID:SCR_023985), to calculate different 2D and 3D properties of the compounds.

### Data analysis

The data analysis modules include data mining and QSAR through various Galaxy-implemented tools, such as Multiconformational Quantitative Structure Activity Relationship (or McQSAR), wekatool, and Support Vector Machine Light (or SVMlight). Expectedly, many computational tools have been developed for drug discovery, and covering all of them is a daunting task and outside the purview of this portal. However, we attempted to integrate the available open-source software packages and link them to our web portal. Here, we briefly describe the selected data analysis tools that may be of high practical utility. The modules include (a) Pharmacophore, (b) Scaffold analysis, (c) Active site analysis, (d) Docking, (e) Screening, (f) Drug repurposing tool, (g) Virtual screening, (h) Visualization, and (i) Sequence alignment.

### Advanced modules

The evolution of the MPDS$^{COVID-19}$ portal is displayed in Figure 5 and Table 1. "Advanced modules" is a new feature, which provides a new dimension to the effectiveness of the MPDS suite in computational drug discovery. It consists of ML algorithms, which can include other tools based on AI, IoT, or large-scale computational tools. A significant number of ML-based tools and methods were developed for drug discovery. Hence, many ML methods and algorithms were incorporated, encompassing different ensemble methods and generalized linear models. Furthermore, the nearest neighbour classification and clustering algorithms were included among the unsupervised ML methods. One of the important features of MPDS$^{COVID-19}$ is 'calculate metrics', which provides parameters for quantifying the quality of the model. This module contains an in-house developed ML-based antiviral prediction model [43]. The detailed procedure for the model generation and evaluation is described in the MPDS$^{COVID-19}$ manual.

### DISCUSSION

The MPDS$^{COVID-19}$ software suite provides important data collected along with molecular modelling and software tools. Besides, the web portal effectively integrates several modules developed in the areas of cheminformatics and CADD, which are available in the Galaxy portal. Our efforts aimed to prepare drug discovery software for the future, focusing on the main challenges ahead. This approach contrasts with conventional ones of developing software with a focus on developing methods and their implementation in an effective way. Our approach warrants an effective integration of AI/ML and IoT-based techniques into molecular modelling, bioinformatics, and cheminformatics approaches. The conventional molecular modelling techniques are based on the principles of classical and quantum mechanics or statistical mechanics. In contrast, data-driven approaches are entirely heuristic and thereby adopt a highly complementary approach compared to molecular modelling. Therefore, effectively integrating these two approaches is indispensable for obtaining deeper insights into drug design problems.



**Figure 5.** Evolution of the MPDS^COVID-19 portal, in terms of time and features, with the integration of new disease-dependent modules (shown in green headers and boxes) and independent modules (shown in red headers and boxes). The MPDS^TB and MPDS^DM were developed in Galaxy 16.01 (released in 2016), while MPDS^COVID-19 was developed with Galaxy 19.05 (released in 2019). There were 35 tools from the Galaxy Toolshed that were integrated into MPDS^TB and MPDS^DM, whereas 59 tools from the Galaxy Toolshed were integrated into MPDS^COVID-19. Compound Library (https://mpds.neist.res.in:8086/), Antiviral Prediction Tool (http://acds.neist.res.in:8501/), and Fragment Library are tools/libraries developed by the group and have been integrated into MPDS^COVID-19. All modules shown in MPDS^COVID-19 are additions to the earlier versions from 2017 and 2018.

## Tips to use MPDS^COVID-19

The MPDS Galaxy user server has been installed at CSIR-NEIST, Jorhat, and can be accessed at [44]. However, valuable data can be obtained, such as drug discovery calculations, retrieval of information, QSAR models, and the generation of workflows. One may refer to the manual available on the website for the usage of several modules and to access the SARS-CoV-2 library. The software suite enables users to submit data from their local computer and freely access and download the outputs. With the authorization of the administrator, users are permitted to alter and include the necessary tools, databases, and software of interest. Despite potential issues with traffic network speed and connectivity (Figure 6), logging into the MPDS server at the host institute might provide helpful information and a comprehensive overview of the MPDS^COVID-19 modules depicted in Figure 7.

A straightforward approach to evaluating the utility of tools or software is to try them with case studies. We provide six case studies and describe them in detail in the supporting information and the manual (Figures S1a to S1f in GitHub [15]). One of the major drawbacks of the MPDS^COVID-19 is that, while the server is freely accessible, it has

**Table 1.** Evolution of drug discovery tools and modules in MPDS portals from MPDS$^{TB}$, MPDS$^{DM}$, to MPDS$^{COVID-19}$.

| Category | Modules | Description | MPDS$^{TB}$ | MPDS$^{DM}$ | MPDS$^{COVID-19}$ |
|---|---|---|:---:|:---:|:---:|
| Data library | Module 1: Literature | Contains disease-related genes, proteins, and polypharmacological information. | √ | √ | √ |
| | Module 2: Target Library | Contains the crystal structures and homology models of disease-related proteins. | √ | √ | √ |
| | Module 3: Disease Library | The disease-related information, analysis and results obtained from our group on a specific disease. | × | × | √ |
| | Module 4: Gene Library | The list of human genes along with the MPDS Aadhar ID | × | √ | √ |
| | Module 5: Compound Library | A single interface window depicting the MPDS Aadhar ID and MPDS Aadhar card of a molecule. | √ | √ | √ |
| | Module 6: Fragment Library | Display 107,614 unique fragments along with their properties. | × | × | √ |
| Data processing | Module 7: File Format Conversion | Conversion of small and macro molecules file format to another file format. Conversion from 2D to 3D using Openbabel. | √ | √ | √ |
| | Module 8: Descriptor Calculation | Calculation of fingerprints and descriptors using CDK and PaDEL tools. | √ | √ | √ |
| Data analysis | Module 9: QSAR | Generation of QSAR models using SVMlight and McQSAR. | √ | √ | √ |
| | Module 10: Pharmacophore | Generation of pharmacophore models using Align-it. | × | × | √ |
| | Module 11: Scaffold analysis | Extracting scaffolds from molecules using strip-it. | × | × | √ |
| | Module 12: Active site analysis | Identification of protein binding pockets using dpocket, fpocket, and Apoc. | × | × | √ |
| | Module 13: Homology modeling | AI-guided 3D structure prediction of a protein using Alphafold. | × | × | √ |
| | Module 14: Docking | Ligand optimization, conformer generation, and molecular docking using Autodock Vina. | √ | √ | √ |
| | Module 15: Screening | Analysing the drug-like features of the molecules to prioritize the compounds using DruLiTo tool; Biopharmaceutical Classification System (BCS); Highlighting the toxicophoric groups in a compound; analysing the natural product likeness of a molecule. | √ | √ | √ |
| | Module 16: Drug repurposing tool | Prediction of biological properties of a molecule using PASS server. | × | √ | √ |
| | Module 17: Virtual screening | Active site definition by creating Frankenstein ligand, cavity definition, molecular docking using rdock. | × | × | √ |
| | Module 18: Visualization | Visualizing protein–ligand interactions using Jmol and Ligplot. | √ | √ | √ |
| Advanced modules | Module 19: Server for antiviral prediction | Machine learning-based prediction of the antiviral nature of a molecule. | × | × | √ |
| | Module 20: Machine learning | Multiple modules for machine learning calculations. | × | × | √ |

limitations in handling multiple jobs. As such, it will be difficult for users to fire jobs that demand substantial computing power. In such cases, we recommend the installation of a local Galaxy MPDS$^{COVID-19}$ server. Nonetheless, the portal is suitable for searching the chemical space and finding information from data library modules.

The disease library modules contain pathways, SARS-CoV-2 protein info, SARS-CoV-2 mutations, host protein information, and literature, which provide factual information about the viral infection, transcription, and immune escape. Also, if a new variant or pandemic emerges, it can be added as a new context. It is worth mentioning that the web portal has tremendous potential to integrate a host of tools, portals, and databases. In addition to the computational modules, users can also explore the "surveillance" and "epidemiology" modules, facilitating the generation of valuable information on the occurrence of pandemics.

## CONCLUSIONS

This work aimed to develop an open-source computational drug discovery platform for COVID-19. The MPDS disease-specific web portal is a customized one-stop solution with

**Figure 6.** The scheme of the architecture of data integration and manual curation, along with the use of an indigenously developed compound library, fragment library, and antiviral prediction machine learning model.

significant potential to drive open-source computational drug discovery. Diseases are different in their pathophysiology and manifestation; therefore, developing a customized approach is fundamental. Especially, in the case of COVID-19, a great deal of data has been generated in a short time owing to the unprecedented impact of the pandemic. The disease-independent modules are valuable for all researchers engaged in drug discovery, especially academics without access to commercial software. In order to make the research more transparent and effective, open-source platforms should emerge as alternatives to commercial computational drug discovery packages [45].

The disease-dependent modules provide a platform for asking profound questions related to a disease. Some questions can be highly focused or provocative: (a) Is it possible to synthesize personalized medicines? (b) Can we understand the altered pathophysiology of a given disease in light of comorbidity, which warrants avoiding/employing specific

**Figure 7.** The diagram enlists different modules available in the MPDS<sup>COVID-19</sup> portal. Galaxy allows the development of workflows by integrating modules to address the problems posed in the area, such as drug discovery. Workflows to address such problems are depicted in some representative case studies (six of them), described in the supporting information hosted in GitHub (Figure S1a–S1f in [15]).

drugs? (c) Can we understand drug–drug interactions and come up with recommendations for administering them in cases where a patient has more than one disease? MPDS is one of the approaches with the potential to formulate and address such questions by augmenting new algorithms and modules.

## AVAILABILITY OF SOURCE CODE AND REQUIREMENTS

- Project name: MPDS-COVID19
- Project homepage: https://mpds.neist.res.in:8085/

- GitHub (Compound Library Module):
  https://github.com/gnsastry/MPDS-Compound_Library
- Documentation: https://mpds.neist.res.in:8085/static/MPDSCOVID-MANUAL.pdf
- Operating system: Platform Independent
- Programming languages: Perl, XML, Python, Bash
- License: GNU GPL v3
- RRID:SCR_023770
- WorkflowHub SEEK ID: https://workflowhub.eu/projects/229

## DATA AVAILABILITY

All data used in this manuscript is publicly available. The data in the 'Data Library' module are available in Zenodo [46].

The supporting information (Tables S1–S4 and Figures S1a–S1f) are freely available at our GitHub repository [15], and the Python codes for the Compound Library Module are freely available and open to others' contributions at
https://github.com/gnsastry/MPDS-Compound_Library. All the data related to the Compound Library are available in Zenodo [47], and snapshots of the code are available in GigaDB [48].

## ABBREVIATIONS

2D, 2 Dimensional; 3D, 3 Dimensional; ADME-Tox, Absorption, Distribution, Metabolism, Excretion, and Toxicity; AI, Artificial Intelligence; CADD, Computer Aided Drug Design; COVID-19, coronavirus disease-19; ID, identifier; InChI, International Chemical Identifier; IoT, Internet of Things; IUPAC, International Union of Pure and Applied Chemistry; KEGG, Kyoto Encyclopaedia of Genes and Genomes; ML, Machine Learning; MPDS, Molecular Property Diagnostic Suite; NCBI, National Center for Biotechnology Information; PDB, Protein Data Bank; PPI, Protein–Protein Interaction; QSAR, Quantitative Structure Activity Relationship; SARS-CoV-2, Severe Acute Respiratory Syndrome-CoronaVirus-2; SMILES, Simplified Molecular-Input Line-Entry System; VOC, Variants Of Concern; VOI, Variants Of Interest; WHO, World Health Organization; XML, Extensible Markup Language.

## DECLARATIONS

### Ethical approval

The authors declare that ethical approval was not required for this type of research.

### Competing interests

The authors declare that they have no competing interests.

### Authors' contributions

GNS conceived the architecture and content of the software. The group members and collaborators of GNS were involved in the development of MPDS for over ten years. All authors have contributed to incorporating the data in the data library and development of the portal. All the authors have proofread the manuscript.

### Funding

This research was financially supported by the Department of Biotechnology (DBT)-Centre of Excellence in Advanced Computation and Data Sciences (Ref. No: BT/PR40188/BTIS/137/27/2021).

## Acknowledgements

DBT is thanked for the financial support in the form of the Centre of Excellence in Advanced Computation and Data Sciences (Ref. No: BT/PR40188/BTIS/137/27/2021). We thank the CSIR for the support and CSIR-NEIST, Jorhat, and CSIR-IICT, Hyderabad, for extending the infrastructure and facilities to carry out the work for the last ten years. GNS thanks all his group members for their inputs and suggestions.

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
