## [Editor Report]

Editor’s AssessmentMPDSCOVID-19 has been developed as a one-stop solution for drug discovery research for COVID-19, running on the Molecular Property Diagnostic Suite (MPDS) platform. This is built upon the open-source Galaxy workflow system, integrating many modules and data specific to COVID-19. Data integrated includes SARS-CoV-2 targets, genes and their pathway information; information on repurposed drugs against various targets of SARS-CoV-2, mutational variants, polypharmacology for COVID-19, drug-drug interaction information, Protein-Protein Interaction (PPI), host protein information, epidemiology, and inhibitors databases. After improvements to the technical description of the platform, testing helped demonstrate the potential to drive open-source computational drug discovery with the platform.

---

## [Reviewer Report]

Comments on revised manuscriptI am satisfied with the changes made to the manuscript and recommend publishing it in its current form if all other reviewers are happy with that.

---

## [Reviewer Report]

Comments on revised manuscriptI have carefully checked the revised work and the author's responses. The authors have made the desired modifications. I have no major concerns on this paper. In the previous review round, Comment #: 3 has not been properly responded by the authors. By data modality, I meant tabular data, graph data, audio data, video data, etc. Authors should add this aspect clearly in the paper about each data modality processed in their system. In Figure 4, some contents (e.g., protein information, PPI interaction, etc.) are unreadable. The abbreviations are not consistently written in terms of small and capital letters. In the paper, the authors are advised to clearly describe the purpose of this tool, who will benefit and in what capacity, why these kinds of tools are needed, etc. I suggest adding such information in abstract to clearly convey the message to readers. In the title, please recheck one word, Open Access or Open Source. The journals are open access while the software are usually open source .

---

## [Reviewer Report]

Reviewer name and names of any other individual's who aided in reviewerPrashanth N SuravajhalaDo you understand and agree to our policy of having open and named reviews, and having your review included with the published manuscript. (If no, please inform the editor that you cannot review this manuscript.)YesIs the language of sufficient quality?YesPlease add additional comments on language quality to clarify if neededIs there a clear statement of need explaining what problems the software is designed to solve and who the target audience is? YesAdditional CommentsThe authors could describe Minimum Information about bioinformatics investigation (MIABI) guidelines
Is the source code available, and has an appropriate Open Source Initiative license <a href="https://opensource.org/licenses" target="_blank">(https://opensource.org/licenses)</a> been assigned to the code?YesAdditional Commentsgithub and Zenodo, yes!  I tested git, forked it and as I didn't test the graphical version, ensured all python libraries are working!
As Open Source Software are there guidelines on how to contribute, report issues or seek support on the code?YesAdditional CommentsYes, pleaseIs the code executable?YesAdditional CommentsIs installation/deployment sufficiently outlined in the paper and documentation, and does it proceed as outlined?YesAdditional CommentsIs the documentation provided clear and user friendly?YesAdditional CommentsYes, a white paper could be more friendly!Is there enough clear information in the documentation to install, run and test this tool, including information on where to seek help if required?YesAdditional CommentsIs there a clearly-stated list of dependencies, and is the core functionality of the software documented to a satisfactory level?YesAdditional Commentsyes with README version!Have any claims of performance been sufficiently tested and compared to other commonly-used packages? YesAdditional CommentsYes, as described by the authorsIs test data available, either included with the submission or openly available via cited third party sources (e.g. accession numbers, data DOIs)?NoAdditional CommentsAre there (ideally real world) examples demonstrating use of the software? YesAdditional CommentsThe Molecular Property Dynamic Suite (MPDS) is a welcome initiative which would serve chemical space for research community. While the authors aimed to deploy it in Galaxy, there is no Galaxy reference cited in first few introductory lines. A strong rationale on Galaxy-MPDS connect could be a value addition The port 8085/8080 are ephemeral and it would be nice if the authors deploy it on a more permanent base An absolute strength for the suite is availability of source code so that end-users can fine tune and reinstantiate the codes. In the architecture, could the end user have a chance to deploy biopython modules for drug discovery/modeling  In Page 4, the authors can define what are the tools precisely used in MPDS 2.3 section: The PPI is not abbreviated as first use The results are exploited well for disease dependent/independent use. However, the major challenge for ligand use/preparation is the use of ncRNAs. Could MPDS provide such instances where ncRNAs could be used fpr targeted ligands? L28 in section 4.1: Pluralis for features ( as one of is used) Also a word or two on aadhar card for perhaps naive users may be mentioned and it may be italicized as it may be a domestic word. Does MPDS suite augur well for Anvaya that Government of India launched, or Tavexa or Taverna? A word to two on local setting up of cloud instance may be a nice addition  Scores on a scale of 0-5 with 5 being the best  Language: 4 Novelty: 4.5 Brevity: 4 Scope and relevance: 4Is automated testing used or are there manual steps described so that the functionality of the software can be verified?YesAdditional Comments Language/Brevity checks:  Page 9 L6: fulfill misspelt webserver are two words, IMHO  Page 10: CADD which IS available  Tabl S2/S4: from THE coronavirdiae space between anticoronavirusdrugs  Figure S3: remove OF (identifying OF existing) Supporting information may be corrected
Any Additional Overall Comments to the Author High resolution Figures esp GA, Figures 2-4 may be insertedRecommendationMinor Revisions

---

## [Reviewer Report]

Reviewer name and names of any other individual's who aided in reviewerAgastya P BhatiDo you understand and agree to our policy of having open and named reviews, and having your review included with the published manuscript. (If no, please inform the editor that you cannot review this manuscript.)YesIs the language of sufficient quality?YesPlease add additional comments on language quality to clarify if neededIs there a clear statement of need explaining what problems the software is designed to solve and who the target audience is? YesAdditional CommentsAs noted in my comments, it would be beneficial to clarify what new capabilities are provided by this new portal over and above what is already available currently.Is the source code available, and has an appropriate Open Source Initiative license <a href="https://opensource.org/licenses" target="_blank">(https://opensource.org/licenses)</a> been assigned to the code?NoAdditional CommentsThere is a github repository (https://github.com/gnsastry/MPDS-18Compound_Library), however, I am unable to access it currently.As Open Source Software are there guidelines on how to contribute, report issues or seek support on the code?YesAdditional CommentsA github repository provides such capabilities. However, it is inaccessible currently.Is the code executable?Unable to testAdditional CommentsIs installation/deployment sufficiently outlined in the paper and documentation, and does it proceed as outlined?Unable to testAdditional CommentsIs the documentation provided clear and user friendly?YesAdditional CommentsIs there enough clear information in the documentation to install, run and test this tool, including information on where to seek help if required?YesAdditional CommentsIs there a clearly-stated list of dependencies, and is the core functionality of the software documented to a satisfactory level?YesAdditional CommentsHave any claims of performance been sufficiently tested and compared to other commonly-used packages? NoAdditional CommentsIs test data available, either included with the submission or openly available via cited third party sources (e.g. accession numbers, data DOIs)?YesAdditional CommentsAre there (ideally real world) examples demonstrating use of the software? YesAdditional CommentsIs automated testing used or are there manual steps described so that the functionality of the software can be verified?NoAdditional CommentsAny Additional Overall Comments to the AuthorMolecular Property Diagnostic Suite for COVID-19 (MPDSCOVID19) is an open-source disease specific web portal aiming to provide a collection of all tools and databases relevant for COVID-19 that are available online along with a few in-house scripts at a single portal. It is built upon another platform called "Galaxy" that provides similar services for data intensive biomedical research. MPDSCOVID19 is in continuation to two other similar disease-specific portals that this group has published earlier - for Tuberculosis and Diabetes mellitus. Overall, MPDSCOVID19 is an interesting and useful resource that could be helpful for biomedical community in conducting COVID-19 related research. It brings together all the databases and relevant tools that may make a researcher's life easier as exemplified through the various case studies included.  I recommend publishing this article after the following revisions noted. Please note that any mention of page numbers below is referring to the reviewer PDF version.   Major revisions:  (1) One main issue in this manuscript is the lack of a clear description of the "new" capabilities provided by MPDSCOVID19 over and above what Galaxy provides. I think a clear distinction between the capabilities/features of Galaxy and MPDSCOVID19 would help improve the manuscript substantially and help readers better understand the capabilities of this new COVID-19 portal.   Further, a description of the additions in the new portal over the earlier TB and Diabetes portals is mentioned on page 7. However, I think more details on such advancements/additions would be beneficial. It could be in the form of a table.  (2) It is mentioned that a major advancement in this new portal is the inclusion of ML/AI models/approaches, however no details have been provided. It would be beneficial to briefly describe what ML based capabilities are included in MPDS and how they can be used by any general user. An additional case study demonstrating the same would be helpful.  (3) MPDS portal provides a collection of tools and databases for COVID-19. However, such resources are ever-growing and hence constant updating of the portal's capabilities/resources would be a necessary requirement for its sustainability. There is no mention of any such plans. Do authors have a modus operandi for the same? Have there been further releases of the previous MPDS portals for TB and Diabetes that may be relevant here?  (4) Page 6 - lines 3-4: I suggest replacing "are going to witness" with "are witnessing". There are several recent advancements in applying ML/AI based approaches to improve different aspects of drug discovery. I recommend including a few references here to this effect. Below are some relevant examples:  (a) 10.1021/acs.jcim.0c00915  (b) 10.1021/acs.jcim.1c00851  (c) 10.1038/s41598-023-28785-9  (d) 10.1098/rsfs.2021.0018  (e) 10.1145/3472456.3473524  (f) 10.1145/3468267.3470573   (5) Page 7 - line 8: I am assuming that the terms like "updates", "additions", etc., used in this paragraph are comparing MPDS with its older versions. If so, it would be beneficial to clarify this explicitly. In addition, I suggest that the authors include a brief literature survey to describe what other tools and/or webservers are available already and how MPDS compares with them. This has not been done so far.  (6) The github repository is currently inaccessible publicly. This needs rectification.  Minor revisions:  (1) Page 4: Before introducing MPDSCOVID19 it makes sense to briefly describe Galaxy and its main features. For instance moving forward lines 19-20 (page 4) and lines 3-6 (page 5) to line 12 (page 4).  (2) Page 5 - line 22: I suggest that authors mention the total number of databases/servers that are covered by MPDSCOVID19 as of now. From Table S1, it appears that there are 15 currently (items 5 and 7 are repeated so the 13 seems the wrong total - needs rectification).  (3) Page 5 - line 30: It would make sense to specify details of the MPDS local server. For instance, how many cores/GPUs are available and what are their hardware architectures? Also, it would be beneficial for the users to know if it is possible to use MPDS tools on their own or public infrastructures for large scale implementations. I suggest authors comment on this aspect too.  (4) Page 6 - lines 16-19: The sentence "Galaxy platform.......extend the availability." needs some rephrasing. It is too long and the hard to comprehend.  (5) Page 7 - line 18: I don't understand the word "colloids". Please clarify.  (6) Page 8 - line 30: "section 2.3" is referred to but I don't see any section numbering the PDF provided. This needs rectification.RecommendationMajor Revisions

---

## [Reviewer Report]

Reviewer name and names of any other individual's who aided in reviewerMAJEED ABDULDo you understand and agree to our policy of having open and named reviews, and having your review included with the published manuscript. (If no, please inform the editor that you cannot review this manuscript.)YesIs the language of sufficient quality?YesPlease add additional comments on language quality to clarify if neededSome changes are needed to make the writing more scientific.Is there a clear statement of need explaining what problems the software is designed to solve and who the target audience is? YesAdditional CommentsIs the source code available, and has an appropriate Open Source Initiative license <a href="https://opensource.org/licenses" target="_blank">(https://opensource.org/licenses)</a> been assigned to the code?YesAdditional CommentsAs Open Source Software are there guidelines on how to contribute, report issues or seek support on the code?YesAdditional CommentsIs the code executable?Unable to testAdditional CommentsIs installation/deployment sufficiently outlined in the paper and documentation, and does it proceed as outlined?Unable to testAdditional CommentsIs the documentation provided clear and user friendly?YesAdditional CommentsIs there enough clear information in the documentation to install, run and test this tool, including information on where to seek help if required?YesAdditional CommentsIs there a clearly-stated list of dependencies, and is the core functionality of the software documented to a satisfactory level?YesAdditional CommentsHave any claims of performance been sufficiently tested and compared to other commonly-used packages? YesAdditional CommentsIs test data available, either included with the submission or openly available via cited third party sources (e.g. accession numbers, data DOIs)?YesAdditional CommentsAre there (ideally real world) examples demonstrating use of the software? YesAdditional CommentsIs automated testing used or are there manual steps described so that the functionality of the software can be verified?YesAdditional CommentsAny Additional Overall Comments to the AuthorIn this paper, the authors introduced a Molecular Property Diagnostic Suite (MPDS), which is a Galaxy-based web portal that was conceived and developed as an open-source disease-specific web portal. MPDS is a customized web portal developed for COVID-19, which is a one-stop solution for drug discovery research. I read the article; it is well-written and well-presented. The enclosed contents can be very useful for researchers working in this field (e.g., COVID-19 systems development). However, I propose some comments/concerns to the current version that need correction during the revision. 1- In the abstract, please provide the technical description of the method’s working. Also, please mention the entities which can benefit from the system.  2- The introduction section doesn’t present the challenges/problems of the existing tools. Please discuss the challenges of the previous such tools and how are they addressed through this new system. 3- I could not find the concrete details of data modalities supported in the system. The authors are advised to include such details. 4- The authors mentioned the use of ML, but I couldn’t find any potential usage of ML models. Please add such analysis during the revision. 5- Also, please add some performance results like time complexity, storage, I/O cost, etc. 6- One comprehensive diagram should be included to better illustrate the working of the proposed tool. 7- Please add limitations of the proposed tool in the revised work. 8- Please add the potential implications of this tool in the context of current/future pandemics.RecommendationMajor Revisions